# CRISPR/Cas13-Mediated Inhibition of EBNA1 for Suppression of Epstein–Barr Virus Transcripts and DNA Load in Nasopharyngeal Carcinoma Cells

**DOI:** 10.3390/v17070899

**Published:** 2025-06-26

**Authors:** Lin Lin, Wai-Yin Lui, Chon Phin Ong, Mabel Yin-Chun Yau, Dong-Yan Jin, Kit-San Yuen

**Affiliations:** 1School of Nursing, Tung Wah College, Kowloon, Hong Kong; 2School of Biomedical Sciences, The University of Hong Kong, Pokfulam, Hong Kong; u3509176@connect.hku.hk (W.-Y.L.); ongcp@hku.hk (C.P.O.); dyjin@hku.hk (D.-Y.J.); 3School of Medical and Health Sciences, Tung Wah College, Kowloon, Hong Kong; mabelyau@twc.edu.hk

**Keywords:** Epstein–Barr virus, nasopharyngeal carcinoma, CRISPR/Cas13, EBNA1 inhibition

## Abstract

Epstein–Barr virus (EBV), a double-stranded DNA virus, is implicated in nasopharyngeal carcinoma (NPC), with particularly high incidence in regions such as southern China and Hong Kong. Although NPC is typically treated with radio- and chemotherapy, outcomes remain poor for advanced-stage diagnoses, highlighting the need for targeted therapies. This study explores the potential of CRISPR/CRISPR-associated protein 13 (Cas13) technology to target essential EBV RNA in NPC cells. Previous research demonstrated that CRISPR/Cas9 could partially reduce EBV load, but suppression was incomplete. Here, the combination of CRISPR/Cas13 with CRISPR/Cas9 shows enhanced viral clearance. Long-term EBNA1 suppression via CRISPR/Cas13 reduced the EBV genome, improved CRISPR/Cas9 effectiveness, and identified suitable AAV serotypes for delivery. Furthermore, cotreatment increased NPC cell sensitivity to 5-fluorouracil and cisplatin. These findings underscore the potential of CRISPR/Cas13 as an anti-EBV therapeutic approach, effectively targeting latent EBV transcripts and complementing existing treatments. The study suggests a promising new direction for developing anti-EBV strategies, potentially benefiting therapies for NPC and other EBV-associated malignancies.

## 1. Introduction

Epstein–Barr virus (EBV) is a double-stranded DNA virus associated with the development of nasopharyngeal carcinoma (NPC) [1]. Over 90% of people worldwide are infected with EBV, but the incidence of NPC varies by region, with higher rates in southern China and Hong Kong [2]. NPC is a head and neck cancer affecting the epithelial lining of the nasopharynx. Although the exact mechanism by which EBV induces oncogenesis in NPC remains unclear, EBV is widely regarded as the primary driver of the disease, as NPC-associated EBV infections are typically monoclonal. The clonality of the EBV episome suggests that NPC tumors originate from a single EBV-infected episome before clonal expansion, reinforcing the idea that latent EBV plays a central role in NPC development [3]. Latent EBV infection may enhance cell proliferation and support host cell survival, contributing to the progression of NPC. NPC is often treated with radio- and chemotherapy, but patients diagnosed at advanced stages may have less favorable outcomes after conventional treatment [4,5]. Therefore, there is a need for a more targeted treatment method to improve clinical outcomes for those with NPC.

EBV typically remains in malignant or transformed cells with more than 10 episomes, and the copy numbers vary among different cell lines. In NPC cells, it is maintained at approximately 20–100 copies [6]. Given that EBV is considered a primary factor in the development of NPC, targeting the EBV episome has emerged as a promising therapeutic approach. Various strategies have been developed in recent years to address this aspect of NPC treatment [7]. The first strategy is to develop small-molecule inhibitors aimed at targeting Epstein–Barr nuclear antigen 1 (EBNA1) in NPC cell lines. Notably, the EBNA1 dimerization inhibitor JLP2 has demonstrated significant efficacy in suppressing the growth of the EBV-positive NPC cell line C666-1 [8]. Likewise, EBNA1 stability is governed by USP7 deubiquitinase and PLOD1 lysine hydroxylase. Inhibitors of either USP7 or PLOD1 destabilize EBNA1 to suppress the proliferation of EBV^+^ tumor cells [9,10]. Furthermore, the peptide-based inhibitor L_2_P_4_ has been shown to bind with EBNA1, selectively inhibiting growth in C666-1 and NPC43 cell lines [11]. Additionally, small-molecule inhibitors VK-1727, VK-1850, and VK-2019 have been found to inhibit EBV tumor growth in xenograft models utilizing NPC cell lines [12] and to decrease EBV DNA levels safely and effectively in phase 1 human clinical trial [13]. The second strategy involves reactivating the EBV lytic cycle using specific agents and combining this with ganciclovir, an antiviral medication used for treating and preventing herpesvirus infection, to selectively eliminate EBV-infected tumor cells [14,15,16]. Another approach involves targeting the EBV genome through precision editing. Utilizing the clustered regularly interspaced short palindromic repeats (CRISPR)/CRISPR-associated protein 9 nuclease (Cas9) system, direct editing of the EBV episome has become possible. Recent studies have indicated that this method can successfully lower the EBV load in infected cells, preventing NPC cell proliferation. Targeting the LMP1 gene using CRISPR/Cas9 results in a significant inhibition of EBV DNA load in human NPC cells and a substantial suppression of LMP1-mediated cell growth [17]. Our previous work also demonstrated that the EBV load in the NPC cell line C666-1 can be reduced by 50% through CRISPR/Cas9 treatment [18].

We previously employed the CRISPR/Cas9 system for editing the EBV genome and performing site-directed mutagenesis [19,20]. In our approach, we used CRISPR/Cas9 to target essential regions of the EBV DNA, including the EBNA1 locus and the origin of plasmid replication (OriP), resulting in a 50% reduction in EBV DNA levels in transfected cells [18]. However, this reduction was not further improved by reintroducing the same guide RNA (gRNA). It remains to be determined whether the gRNA binding sites in some EBV clones might be mutated to evade CRISPR/Cas9-mediated DNA editing. These findings demonstrate that while CRISPR/Cas9 can reduce EBV DNA, it is not sufficient to completely eradicate latent EBV infections in NPC cells. Interestingly, although the survival of C666-1 cells was not impacted by EBV suppression via CRISPR/Cas9 treatment, the cells became more sensitive to the chemotherapeutic agents cisplatin and 5-fluorouracil. Overall, these findings serve as a proof-of-concept for using CRISPR/Cas9 to reduce EBV DNA in NPC cells and suggest a potential strategy for increasing the sensitivity of EBV-infected NPC cells to chemotherapy. Further investigation and optimization is needed to enhance the translational potential of these innovative therapeutic approaches.

Since the discovery of the canonical CRISPR/Cas9 molecules, the CRISPR/Cas system has been extensively studied. This research has led to the development of a novel RNA-targeting CRISPR enzyme, CRISPR-associated protein 13 (Cas13), based on computational modeling [21,22,23]. Cas13 is a type VI CRISPR nuclease that exists in four different subtypes: Cas13a, Cas13b, Cas13c, and Cas13d. All Cas13 molecules can bind and cleave single-stranded RNA using a gRNA. This gRNA is approximately 64 nucleotides long and forms a complex with the Cas13 nuclease via a short hairpin in the CRISPR RNA (crRNA). Target specificity is achieved by a 28–30 base-pair (bp) spacer that is complementary to the target region. In bacteria, after a Cas13 molecule recognizes and cleaves its target transcript, it induces the non-specific degradation of nearby RNA transcripts, in addition to the execution of its programmable RNase activity. However, Cas13 molecules do not exhibit this non-specific RNA degradation activity in mammalian cells, which allows CRISPR/Cas13 systems to be accurately deployed in human cells for specific transcript knockdown [21,22,23]. Among the four types of Cas13, Cas13d (also known as RfxCas13d) has the most robust knockdown activity in mammalian cells and can effectively degrade endogenous transcripts [21]. The CRISPR/Cas13 system was subsequently applied to the field of virology, particularly for inhibiting RNA viruses such as SARS-CoV-2, lymphocytic choriomeningitis virus (LCMV), influenza A virus (IAV), vesicular stomatitis virus (VSV), and enterovirus [24,25,26], as well as for detecting various viruses, including EBV [27,28]. These studies have demonstrated that the CRISPR/Cas13 system is promising for suppressing the replication of RNA viruses and can serve as a rapid detection method for viral genomes. However, the potential use of CRISPR/Cas13 for inhibiting DNA viruses remains unexplored. For DNA viruses, CRISPR/Cas9 offers a more direct approach and has significant potential to target and eliminate viral DNA genomes within infected cells. However, as previously mentioned, the repair mechanisms induced by CRISPR/Cas9 can enable the virus to escape elimination, allowing some viral residues to persist in the cells. Therefore, exploring alternative strategies for targeting DNA viruses is valuable. Since CRISPR/Cas13 can effectively knock down specific transcripts in human cells, it may also be capable of targeting essential viral transcripts, thereby reducing viral activity and maintenance within infected cells. We propose that using CRISPR/Cas13 to target key transcripts of EBV could be a promising method to disrupt EBV episome maintenance. Additionally, it may serve as an adjunct method to complement other anti-EBV therapies.

In this study, we aimed to assess the anti-EBV potential of CRISPR/Cas13 in NPC cells. We targeted various important EBV transcripts, including the episome maintenance protein EBNA1, the key oncogenic proteins latent membrane proteins 1 and 2 (LMP1 and LMP2), and the major lytic activators Zta and Rta, for knockdown using CRISPR/Cas13. We documented the knockdown efficiency, as well as its impact on EBV load, cell viability and susceptibility to chemotherapy. Additionally, we evaluated the synergistic effects with CRISPR/Cas9 strategies and recorded the efficiency of different serotypes of adeno-associated virus (AAV) vectors for delivering CRISPR/Cas systems in nasopharyngeal epithelial (NPE) and NPC cells. Our research demonstrated the potential of CRISPR-Cas13-mediated EBV transcript knockdown as an anti-EBV therapy and as an adjunct to existing methods. Importantly, this research provides the first evidence that CRISPR/Cas13 can effectively suppress EBV in NPC and may also be applicable to other DNA virus infections, highlighting its promise as a therapeutic strategy for NPC.

## 2. Materials and Methods

### 2.1. Plasmids and crRNA Design

The CRISPR/Cas13 plasmids PXR001 (RfxCas13d expression construct) and PXR003 (human U6-driven expression vector for crRNAs compatible with CasRx) were obtained from Patrick Hsu laboratory [21] through Addgene (IDs: 109049 and 109053). The crRNAs were designed using RNAfold, selecting sequences with minimal secondary structure—defined as one or fewer possible secondary structures with a minimal free energy greater than −0.1 kcal/mol. These sequences were subsequently analyzed using the Basic Local Alignment Search Tool (BLAST) against the human transcriptome to identify any potential off-target binding. crRNA sequences that matched with one or fewer potential human targets were chosen for plasmid construction. The CRISPR crRNA expression constructs were created by inserting the respective crRNA sequences into PXR003. The crRNA sequences were as follows:

EBNA1-crRNA-1: 5′-CATCTCTATGTCTTGGCCCTGAT-3′.

EBNA1-crRNA-2: 5′-TATTCAAAATAATCGGCATCCCC-3′.

EBNA1-crRNA-3: 5′-ACGATGCTTTCCAAACCACCCTC-3′.

EBNA1-crRNA-4: 5′-ACCTCCATATACGAACACACCGG-3′.

EBNA1-crRNA-5: 5′-GAAATAACAGACAATGGACTCCC-3′.

LMP1-crRNA-1: 5′-ACTCATAACGATATACAGCCAGA-3′.

LMP1-crRNA-2: 5′-CCGTGCAAATTCCAGAGAGCGAT-3′.

LMP1-crRNA-3: 5′-ACCTAAGACAAGTAAGCAGCCAA-3′.

LMP1-crRNA-4: 5′-CAGCCAAAACAGTAGCGCCAAGA-3′.

LMP2-crRNA-1: 5′-ATTGCAAAGAAAGTGACAACCGC-3′.

LMP2-crRNA-2: 5′-GAAACCACAGTTACAGCTCCAAG-3′.

LMP2-crRNA-3: 5′-CCCAAAATCAGTGACGCTAGCAG-3′.

Rta-crRNA-1: 5′-GCCTCAGAAAGTCTTCCAAGCCA-3′.

Rta-crRNA-2: 5′-CAAAAATTGCAGATGTTGAGCGT-3′.

Rta-crRNA-3: 5′-AACACGTCAGATATGGCACTGCC-3′.

Zta-crRNA-1: 5′-AAAAGCTTGTACAAAAGGCACCT-3′.

Zta-crRNA-2: 5′-GCACACAAGGCAAAGGAGCTTGT-3′.

Zta-crRNA-3: 5′-GAAACATGATAGGCAGTGAGCTG-3′.

NC-crRNA (Negative control): 5′-TCACCAGAAGCGTACCATACTCA-3′.

The CRISPR/Cas9 expression constructs PX459-EBNA1-gRNA1, PX459-EBNA1-gRNA-2, and PX459-OriP-gRNA-1 were created as described [18].

### 2.2. Cell Culture and Transfection

The nasopharyngeal carcinoma cell line C666-1 was cultured in RPMI 1640 medium (GIBCO, Life Technologies, Carlsbad, CA, USA) supplemented with 10% fetal bovine serum (FBS; GIBCO, Life Technologies). The nasopharyngeal carcinoma cell line NPC43 was cultured in RPMI 1640 supplemented with 10% FBS and 0.2% Rho kinase inhibitor Y-27632 (ROCK inhibitor; Tocris Bioscience, Bristol, UK). The nasopharyngeal epithelial cell line NP460 was maintained in a medium containing a 1:1 mixture of EpiLife^®^ medium (Gibco, Waltham, MA, USA) and 1% EpiLife^®^ defined growth supplement (EDGS; Gibco). Transfection of C666-1 was performed using the TransIT^®^-Keratinocyte Transfection Reagent (Mirus, Marietta, GA, USA) and the transfection of NPC43 was performed using the jetPRIME^®^ transfection reagent (Polyplus, Illkirch-Graffenstaden, France).

### 2.3. qRT-PCR Analysis

Total RNA was extracted using FreeZol Reagent (Vazyme, Nanjing, China), and cDNA was synthesized with the PrimeScript RT Reagent Kit with gDNA Eraser (Takara, Kusatsu Japan). Genomic DNA contamination was eliminated by treating the RNA with gDNA Eraser for 2 min at 42 °C, followed by cDNA synthesis using PrimeScript RT Enzyme for 15 min at 37 °C. The RT enzyme was then heat-inactivated for 5 s at 85 °C. The transcript levels of EBNA1, LMP1, LMP2, Rta, and Zta were determined by ChamQ SYBR Color qPCR Master Mix (Vazyme) in StepOne Real-Time PCR System (Thermo Fisher, Waltham, MA, USA). The transcript levels were normalized with tubulin transcript level. The sequences of qPCR primers were as follows:

For EBNA1 targeting using EBNA1-crRNA-1:

F: 5′-TAACCATGGACGAGGACGGG-3′.

R: 5′-TGCAACTTGGACGTTTTTGGG-3′.

For EBNA1 targeting using EBNA1-crRNA-2 and 3:

F: 5′-AGGCCATTTTTCCACCCTGT-3′.

R: 5′-TTGGAACCTCCTTGACCACG-3′.

For EBNA1 targeting using EBNA1-crRNA-4 and 5:

F: 5′-CCACAATGTCGTATTACACC-3′.

R: 5′-ATAACAGACAATGGACTCCCT-3′.

For LMP1 targeting using LMP1-crRNA-1 and 4:

F: 5′-TCCATAGGCCTTGCTCTCCT-3′.

R: 5′-CGCTCCTCCAGTCCAGTTAC-3′.

For LMP1 targeting using LMP1-crRNA-2 and 3:

F: 5′-GGAGGCCTTGGTCTACTCCT-3′.

R: 5′-CACCAAGCCGCCAGAGAATC-3′.

For LMP2:

F: 5′-TGGCGGACTACAAGGCATTT-3′.

R: 5′-AACATCATGCCGCCACAAAC-3′.

For Rta targeting using Rta-crRNA-1 and 2:

F: 5′-ATGAGGCCTAAAAAGGATGGC-3′.

R: 5′-TGTTTCGGAGAATGGCCCAA-3′.

For Rta targeting using Rta-crRNA-3:

F: 5′-GTATGTTCCTGCCAAAGCCG-3′.

R: 5′-TTTGGCTGACACACCTCTCG-3′.

For Zta:

F: 5′-TACACCTGACCCATACCAGG-3′.

R: 5′-CAGGAAACCACGACCCAGTT-3′.

For tubulin:

F: 5′-AAGATCCGAGAAGAATACCCTGA-3′.

R: 5′-CTACCAACTGATGGACGGAGA-3′.

### 2.4. EBV DNA Quantification

EBV DNA quantification was performed as described [18]. The total genomic DNA was purified using a Quick-DNA™ Miniprep Plus Kit (Zymo Research, Irvine, CA, USA). DNA was diluted 1000-fold for quantitative PCR (qPCR) analysis. TaqMan^®^ probes detecting the EBV genome and GAPDH gene (probe Hs02758991_g1) were ordered from Applied Biosystems, and the TaqMan^®^ universal PCR Master Mix (Applied Biosystems) was used in the PCR reactions. qPCR was carried out in the StepOnePlus™ Real-Time PCR System (Applied Biosystems, Waltham, MA, USA) using 40 cycles of amplification (95 °C for 10 s and 60 °C for 1 min). Each sample was measured in triplicate. Primers for EBV were 5′-ACCACGACACACTGATGAACAC-3′ (forward) and 5′- CTAGAATCGTCGGTAGCTTGTTGA-3′ (reverse). Amplicon size was 72 bp. Probe for EBV was 6FAM-ACTCCCTCCCGCACCC-MGB. The levels of EBV DNA relative to genomic GAPDH DNA were calculated from 2^−ΔΔCT^ using the comparative CT method [29].

### 2.5. Western Blotting

Protein detection was performed as described [30]. Cells were lysed in RIPA buffer supplemented with a protease inhibitor. Each well of a 6-well plate received 200 μL of RIPA buffer for a 30 min incubation at 4 °C. Cell lysates were collected by centrifugation at 13,500 rpm for 15 min at 4 °C. Protein concentrations were measured using the Bradford assay. Protein samples were boiled for 10 min with protein sample buffer and subjected to SDS-PAGE analysis using a 10% discontinuous polyacrylamide gel. Following electrophoresis, proteins in the SDS-PAGE gel were transferred to an Immobilon-P polyvinylidene difluoride (PVDF) membrane (Millipore, Burlington, MA, USA) via electroblotting, then blocked with 5% skim milk in TBST buffer for 1 h. After blocking, the primary antibody was applied in 5% skim milk/TBST overnight at 4 °C, followed by the corresponding secondary antibody at room temperature for 1 h the next day. Proteins were visualized with the ECL system (GE Healthcare Life Science, Chicago, IL, USA). EBNA1 was detected using the primary antibody EBV EBNA-1 Antibody (1EB12) (sc-81581: Santa Cruz, Santa Cruz, CA, USA), and GAPDH was detected using GAPDH Antibody (G-9) (sc-365062: Santa Cruz).

### 2.6. AAV Serotyping and Infection Assay

AAV serotyping was performed using an AAVPrime™ AAV Serotype Testing Kit Plus (GeneCopoeia, Rockville, MD, USA). Twelve premade GFP-expressing AAV serotypes (1, 2, 3, 4, 5, 6, 7, 8, 9, 10, DJ, and DJ/8) were used for serotyping. C666-1, NPC43, and NP460 cells were seeded in a 24-well plate and infected at multiplicities of infection (MOI) of 1:100, 1:1000, and 1:10,000 for the AAV serotypes. The cells were pretreated with 0.8 µM camptothecin (MCE) in the medium for 4 h before the infection. The viral stocks were diluted to the required MOI in 200 µL of culture medium supplemented with 2% (*v*/*v*) heat-inactivated FBS. Two hundred microliters of diluted virus was added to the cells and incubated at 37 °C in a CO_2_ incubator for 2 h. Following incubation, 200 µL of pre-warmed culture medium containing 18% (*v*/*v*) heat-inactivated FBS was added to the infected cells, and the GFP signal was analyzed 48 h post-infection. The GFP signal was captured by fluorescence microscopy using a Nikon inverted microscope and quantified by RT-qPCR with the primers 5′-AGAACGGCATCAAGGTGAAC-3′ (forward) and 5′-TGCTCAGGTAGTGGTTGTCG-3′ (reverse). The recombinant serotype 2 AAV expressing EBNA1-crRNA-5 and a non-target control (NC-crRNA) in the pAV-U6-gRNA-CMV-nls-CasRx vector (WZ Biosciences Inc., Rockville, MD, USA) was constructed and packaged by WZ Biosciences Inc. The virus was used to infect cells at an MOI of 1:1000.

### 2.7. Cell Viability Assay

The viability of C666-1 cells was assessed using the MTS Cell Proliferation Assay Kit (Colorimetric, Abcam, Cambridge, UK). A total of 5 × 10^4^ C666-1 cells were placed in a 96-well plate with 200 μL of growth medium one day before the addition of the MTS reagent. Then, 20 μL of MTS reagent was added to each well, and the cells were incubated for 4 h at 37 °C under standard culture conditions. The spectrophotometric absorbance of the samples at 490 nm was measured using a microplate reader, with higher absorbance indicating greater cell viability. For CRISPR-treated groups, cells were edited with specific CRISPR constructs and then exposed to 0.1 mM of cisplatin for 4 h or 1 μM of 5-fluorouracil for 24 h in order to assess the viability of CRISPR-treated cells compared to untreated controls.

### 2.8. Statistical Analysis

All tests were performed with three biological replicates, and statistical significance was assessed using two-tailed paired Student t test. The results are expressed as mean ± standard deviation (SD). Significance is indicated in the figures by asterisks (ns: *p* > 0.05; *: *p* < 0.05; **: *p* < 0.01; ***: *p* < 0.001).

## 3. Results

### 3.1. CRISPR/Cas13-Mediated Suppression of EBV Transcripts in Nasopharyngeal Carcinoma Cells

C666-1 is an NPC cell line that naturally carries EBV and exhibits the EBV latency II pattern. It expresses EBV-encoded RNAs (EBER I and II), the EBNA1 protein, LMP1 and LMP2 transcripts, and miR-BARTs [31,32]. However, it cannot complete the full lytic replication cycle or produce infectious virions due to a nonsense mutation in the lytic gene BNRF1, which encodes a major tegument protein of EBV [33]. It serves as a good model for NPC latently infected with EBV. In our previous study, we demonstrated that using CRISPR/Cas9 to target essential regions can reduce its EBV load [18]. Therefore, C666-1 was used in our study to assess the feasibility of CRISPR/Cas13 in suppressing the EBV transcripts.

Five EBV transcripts, EBNA1, LMP1, LMP2, Zta, and Rta, were chosen as CRISPR/Cas13 targets. EBNA1, LMP1, and LMP2 are involved in the EBV type II latency program and are regarded as important regulators of NPC pathogenesis. EBNA1 is essential for retaining the EBV episome in latently infected cells, and thus, downregulating EBNA1 results in a significant loss of the EBV genome from cells [12]. LMP1 and LMP2 are key oncogenic proteins of EBV, possessing many oncogenic properties, and they contribute to chemoresistance in some NPC cells [34]. Therefore, downregulating LMP1 and LMP2 may help sensitize NPC cells to chemotherapeutics. Although Zta and Rta are not usually expressed during type II latency, treatment with chemotherapeutics such as cisplatin and 5-fluorouracil (5-FU) can trigger Zta and Rta expression, thereby reactivating lytic replication [35]. Some lytic proteins, such as BamHI rightward reading frame 1 and BamHI A leftward fragment 1, can help EBV-infected cells evade apoptosis [36], and thus, these proteins may interfere with the cytotoxic effects of chemotherapeutics, with their inducers Zta and Rta potentially contributing to chemoresistance. Although C666-1 cannot complete the full lytic replication cycle, it is still capable of expressing early lytic transcripts, including Zta and Rta, when induced by various agents [37,38]. Therefore, C666-1 can also be used to examine the effects of targeting Rta and Zta with CRISPR/Cas13 under appropriate induction conditions.

The crRNAs targeting EBV transcripts were designed using RNAfold (http://rna.tbi.univie.ac.at/cgi-bin/RNAWebSuite/RNAfold.cgi, accessed on 29 May 2023), an online computational tool that predicts the secondary structures of single-stranded DNA or RNA sequences [39]. Regions with minimal secondary structure and sequences conserved across common EBV strains were primarily selected. Since CRISPR/Cas13 recognizes target sites via simple base pairing with a 28–30 bp gRNA, there is a risk of non-specific gRNA binding, which can lead to off-target effects and unintended transcript degradation [23]. To minimize these off-target effects, the designed crRNAs were analyzed against the human transcriptome using the Basic Local Alignment Search Tool (BLAST) (https://blast.ncbi.nlm.nih.gov/Blast.cgi, accessed on 29 May 2023) to identify any potential off-target binding. Those designs that showed a high potential for matching human transcripts were eliminated. The designed crRNAs were subsequently cloned into the crRNA expression plasmid pXR003, a human U6-driven vector compatible with CasRx, to facilitate expression in mammalian cells [21]. There are five crRNAs for EBNA1, four for LMP1, three for LMP2, three for Zta, and three for Rta that were designed and constructed for this study. Their target regions are illustrated in Figure 1.

To assess the knockdown efficiency of the crRNAs, we transfected C666-1 cells with the CasRx expression plasmid (pXR001) and pXR003-EBNA1/LMP1/LMP2/Rta/Zta crRNA over a 5-day period, conducting two rounds of transfection. A CasRx (pXR001)-to-crRNA (pXR003) ratio of 1:5 was used. The knockdown efficiency of each crRNA was determined by RT-qPCR with specific primers. Most detection primers were designed to flank the target cut site of the crRNA, minimizing the possibility that Cas13-cleaved RNA remaining in the RNA extraction pool could contribute to cDNA synthesis and lead to false-positive detection of the uncleaved target transcript (Figure 1). LMP1 and EBNA1 are expressed in C666-1 cells during latency and were detected directly following transfection (Figure 2A,B). Zta and Rta, which are key lytic activators of EBV, are not expressed in latency II cells. To evaluate the knockdown efficiency of Zta and Rta crRNAs, the crRNA expression plasmids were first transfected into the cells. Zta and Rta were then induced using 1.5 mM sodium butyrate 24 h prior to RNA harvesting (Figure 2C,D). Because LMP2 expression is low and challenging to detect in C666-1 cells, the newly established EBV-carrying NPC cell line, NPC43, was used for LMP2 detection [40]. Although LMP2 expression is barely detectable in NPC43, it appears that none of the crRNA designs effectively knocked it down (Figure 2E). Among all the crRNA designs, six crRNAs were able to significantly reduce the target transcripts by more than 50% efficiency, as determined by Student’s *t*-test. These include EBNA1-crRNA-3 (*p* = 0.005), EBNA1-crRNA-5 (*p* = 0.004), LMP1-crRNA-3 (*p* = 0.017), Rta-crRNA-1 (*p* = 0.001), Zta-crRNA-2 (*p* = 0.004), and Zta-crRNA-3 (*p* = 0.008). Apart from LMP2, which is barely detectable in our cell model, the remaining four target transcripts were successfully knocked down by more than 50% using CRISPR/Cas13. The knockdown efficiency of EBNA1 was also confirmed by Western blotting, which showed that the protein level was reduced by about half following treatment with EBNA-crRNA-2, EBNA-crRNA-3, and EBNA-crRNA-5 (Figure 2F). Since the CRISPR/Cas13 constructs are delivered via co-transfection, the reduction in target EBV transcripts may be limited by transfection efficiency. The knockdown efficiency could potentially be enhanced by employing more effective delivery methods, such as using viral vectors.

### 3.2. CRISPR/Cas13-Mediated Suppression of EBV Transcripts Reduces the Viral DNA Load in C666-1 Cells

The selected crRNAs demonstrated over 50% knockdown efficiency of target EBV transcripts, prompting further investigation into their ability to affect latent EBV load in C666-1 cells. After a short-term transfection of 5 days, suppression of the target EBV transcripts was observed (Figure 2), but there was no significant reduction in the EBV load (Figure 3A). However, with extended transfection lasting at least two weeks, both EBNA1-targeting crRNAs (EBNA1-crRNA-3 and EBNA1-crRNA-5) led to a significant decrease in EBV load. EBNA1-crRNA-5 exhibited the most pronounced effect, reducing the EBV load by 32% at week 2 and 38% at week 3 (Figure 3A). In contrast, the crRNAs targeting LMP1, Zta, and Rta did not significantly impact the EBV load, even with long-term treatment. This observation is not very surprising, as these transcripts, while crucial in the EBV life cycle, do not play a vital role in maintaining the EBV genome within latently infected cells as EBNA1 does. While knocking down these transcripts may influence EBV activity within infected cells, it may be insufficient to eliminate the EBV episome. Targeting EBNA1 is the most effective approach because it is the primary protein responsible for episome maintenance. Its effects take time to appear and become noticeable after about two weeks. This delay is likely due to EBNA1’s relatively long half-life [41], which means existing proteins need time to degrade before impacting the EBV episome. Although CRISPR/Cas13 does not completely knock down the EBNA1 transcript, it still achieves an about 40% reduction in EBV load with long-term treatment. This demonstrates that the strategy of CRISPR/Cas13-mediated suppression of EBNA1 effectively reduces the EBV load in NPC cells.

### 3.3. CRISPR/Cas13-Mediated Suppression of EBNA1 Transcripts Sensitized C666-1 Cells to CRISPR/Cas9-Mediated Elimination of the EBV Genome

In addition to the direct anti-EBV effects of CRISPR/Cas13-mediated suppression of EBV transcripts in NPC cells, its potential as an adjuvant therapy was also assessed. Previously, we demonstrated that CRISPR/Cas9 targeting of EBNA1 and OriP can reduce EBV DNA levels by 50% in transfected cells [18]. However, reintroducing the same gRNA does not further increase this reduction. Given that CRISPR/Cas9 and Cas13 have similar mechanisms and can be delivered in the same manner, the CRISPR/Cas13 system might serve as an effective adjuvant strategy to enhance the effects of the CRISPR/Cas9 system. To test this, C666-1 cells were co-transfected with our previously constructed gRNA-Cas9 co-expression plasmids targeting the EBNA1 and OriP regions (PX459-EBNA1-gRNA1, PX459-EBNA1-gRNA-2, and PX459-OriP-gRNA-1) [18] along with the CRISPR/Cas13 expression constructs at a 1:4 ratio.

The representative crRNAs targeting EBNA1 (EBNA1-crRNA-5), LMP1 (LMP1-crRNA-3), and Zta (Zta-crRNA-2) was tested for their adjuvant ability. Since both Zta and Rta serve as lytic inducers with similar functions, Zta was selected for testing because it acts as an upstream activator of Rta [42]. A non-relevant crRNA (NC-crRNA) was used as a negative control for comparison. Upon co-transfection, the NC-crRNA group treated with CRISPR/Cas9 demonstrated approximately a 40% reduction in EBV load after about two weeks of treatment, reflecting the suppressive effect of the CRISPR/Cas9 system we previously reported. With the addition of EBV-crRNAs and CasRx, suppressing EBNA1 with EBNA-1-crRNA-5 further reduced the EBV load from about 55% to 29% under CRISPR/Cas9 treatment (Figure 3B). This suggests that the suppression of EBNA1 transcripts by CRISPR/Cas13 can sensitize C666-1 cells to more effectively lose EBV through CRISPR/Cas9-mediated cleavage. This may be due to the reduction in EBNA1 levels decreasing the binding of the EBV episome, making the damaged EBV genome more likely to be lost rather than repaired by DNA repair mechanisms, or facilitating the elimination of defective EBV by reducing support from the EBNA1 protein. However, targeting Zta and LMP1 does not seem to have any significant effects on CRISPR/Cas9-mediated reduction in EBV load (Figure 3B). This may be because their effects on the EBV episome are more indirect and may only become apparent during prolonged treatment, which cannot currently be achieved with the multiple transfection system.

### 3.4. Transduction Efficiency of Various Adeno-Associated Virus (AAV) Serotypes in NPE and NPC Cells

Although CRISPR/Cas13 has shown potential in anti-EBV strategies and as an adjuvant to other therapeutic methods, the need for multiple transfections limits its functional effectiveness. Additionally, for translational purposes, effective delivery strategies for the CRISPR/Cas13 system need to be developed for its clinical application. Viral delivery vectors, including adeno-associated viruses (AAV), adenoviral vectors (AdV), and lentiviral vectors (LV), are the most commonly used systems to deliver the CRISPR/Cas system in vivo [43]. AAVs have emerged as the leading choice among viral vectors, attributed to their favorable safety profile and therapeutic efficacy. AAVs are distinguished by their low immunogenicity relative to other viral vectors [44,45], enhancing their appeal for clinical applications. Post-transduction, AAV DNA predominantly exists in an episomal form, with integration occurring at specific loci, including mitochondrial DNA hotspots and the AAVS1 site on chromosome 19 [46,47,48]. This precise integration contrasts with the random insertion characteristic of lentiviral vectors. Crucially, these integration sites are considered safe and do not facilitate tumorigenesis, underscoring the reliability and advantages of AAV in gene therapy. In light of this, AAV is the leading platform for in vivo gene therapy and has been widely used for an efficient delivery of CRISPR/Cas9 and CRISPR/Cas13 systems into target cells [49,50]. AAV is characterized by its existence in various serotypes with different tropisms. Given that the susceptibility of NPE and NPC cell lines to various AAV serotypes infection is unclear, we examined the transduction efficiency of various AAV serotypes in C666-1, NPC43, and NP460 cells to determine the best serotype for application to nasopharyngeal cells. Pre-made green fluorescent protein (GFP)-expressing AAV in serotypes 1, 2, 3, 4, 5, 6, 7, 8, 9, 10, DJ and DJ/8 were obtained from the AAVPrime™ Adeno-associated Virus Serotype Testing Kit (GeneCopoeia) and were used to infect the NPC and NPE cell lines. The transduction efficiency was assessed by observing the GFP signal through fluorescence microscopy (Appendix A) and the GFP expression level was quantified using qRT-PCR (Figure 4A–C). Forty-eight hours after transduction at an MOI of 1:10,000, AAV 6 and AAV D/J exhibited the highest infection efficiencies in the NPE cell line NP460 (Figure 4A). For the NPC cell lines, C666-1 and NPC43, AAV 2, 3, 6, 9, and D/J displayed transduction efficiency, with AAV 2 and AAV D/J being the most effective in both C666-1 and NPC43 cells (Figure 4B,C). It is intriguing that NPC and NPE cells exhibit different tropisms for AAV serotypes. AAV6 shows a strong effect on normal NPE cells but only moderate efficiency in NPC cell lines, suggesting that different AAV serotypes might be needed to effectively target NPE and NPC cells. This discrepancy may be due to differences in cell surface molecules between NPE and NPC cells. Since AAV6 primarily binds to glycans containing sialic acid residues on glycoproteins and glycolipids on the cell membrane [51], changes in glycosylation patterns associated with NPC oncogenesis could affect AAV6 infectivity. Further investigation is warranted to determine whether these glycosylation changes are influenced by EBV [52,53]. In summary, specific AAV serotypes can effectively transduce both NPE and NPC cells, making them suitable for the efficient delivery of CRISPR constructs.

### 3.5. Enhancing Chemotherapeutic Sensitivity in C666-1 Cells Through CRISPR/Cas13 Targeting of EBNA1

In addition to the CRISPR/Cas9 system’s ability to enhance the editing of the EBV genome, we investigated the potential of CRISPR/Cas13-mediated EBV suppression to augment the effects of traditional chemotherapeutic treatments. To assess this, we measured cell viability following chemotherapeutic treatments using the MTS assay. Previously, we reported that the CRISPR/Cas9 system can increase the sensitivity of C666-1 cells to chemotherapeutic agents [18], so we used the CRISPR/Cas9 system as a control in our experiments. Furthermore, we examined whether there are any additive effects between the CRISPR/Cas9 and Cas13 strategies in enhancing chemotherapeutic efficacy. To improve the delivery efficiency of the CRISPR/Cas13 system in C666-1 cells, a recombinant serotype 2 AAV expressing EBNA1-crRNA-5 and a non-target control (NC-crRNA) was constructed using the pAV-U6-gRNA-CMV-nls-CasRx vector and packaged by WZ Biosciences Inc. AAV2 was selected due to its superior performance in C666-1 cells compared to other AAV serotypes (Figure 4C). The cells were transduced with either EBNA1-crRNA-5/CasRx co-expressing AAV2 (EBNA1-crRNA-5 AAV2) or NC-crRNA/CasRx co-expressing AAV2 (NC-crRNA AAV2) and were then transfected with CRISPR/Cas9 constructs targeting the EBNA1 and OriP regions (PX459-EBNA1-gRNAs + PX459-OriP gRNA) or a mock vector (PX459). This process was repeated over approximately a month, and the treated cells were assessed using the MTS assay.

There was no significant change in cell viability in groups treated with Cas9, Cas13, or both (Figure 5, lanes 1 to 4). However, cell viability was significantly reduced in the Cas13-EBNA1-crRNA 5-transduced group compared to the Cas13-NC-crRNA group after 5-fluorouracil treatment, dropping from 79.6% to 66.7% (*p* = 0.046) (Figure 5, lane 9 compared to lane 11). A similar pattern was observed with cisplatin treatment, but the difference was not statistically significant (*p* = 0.18) (Figure 5, lane 5 compared to lane 7), likely due to the strong cytotoxic effects of cisplatin causing high variability in viability measurements. The reduction in cell viability with 5-fluorouracil suggests that CRISPR/Cas13-mediated suppression of EBNA1 can sensitize cells to chemotherapeutic agents. Additionally, cell viability decreased from 34.8% to 21.1% under cisplatin treatment (Figure 5, lane 10 compared to 12) and from 51.1% to 30.9% under 5-fluorouracil treatment (Figure 5, lane 6 compared to 8) when comparing the CRISPR/Cas9 single-treated group to the group co-treated with CRISPR/Cas9 and Cas13 (*p* = 0.041 and *p* = 0.003, respectively). This indicates that targeting EBNA1 with CRISPR/Cas13 can enhance the chemotherapeutic effects mediated by CRISPR/Cas9 cleavage.

## 4. Discussion

In this study, we demonstrated that CRISPR/Cas13-mediated suppression of EBNA1 can effectively reduce the EBV genome in NPC cells. CRISPR/Cas13 was able to downregulate both latent and lytic transcripts, including EBNA1, LMP1, Zta, and Rta, in EBV-infected cells (Figure 2). Long-term suppression of EBNA1 using CRISPR/Cas13 not only showed potential for reducing the EBV genome in NPC cells (Figure 3A) but also enhanced the effectiveness of CRISPR/Cas9-mediated EBV genome removal in NPC cells (Figure 3B). We identified the most suitable AAV serotypes for CRISPR/Cas delivery to NPC and NPE cells, with AAV6 and AAV D/J being most effective in NPE, and AAV2 and AAV D/J in NPC cell lines (Figure 4). Targeting EBNA1 with CRISPR/Cas13 and co-treatment with CRISPR/Cas9 were also shown to enhance chemotherapeutic sensitivity in C666-1 cells, particularly to 5-fluorouracil and cisplatin (Figure 5). Taken together, our study presents a proof-of-concept that CRISPR/Cas13-mediated suppression of EBV transcripts can effectively target latent EBV transcripts and serve as an adjunct to other anti-EBV or NPC therapies, such as CRISPR/Cas9-mediated EBV editing and chemotherapy. This approach may offer a new direction for developing anti-EBV strategies for treating NPC and other EBV-associated carcinomas.

Our evaluation of various targets revealed that EBNA1 offers the greatest potential for anti-EBV therapy. This protein is vital for the EBV life cycle, as it ensures the maintenance of the viral episome in latently infected cells. EBNA1 is present in all latency states of EBV-associated tumors, making it an ideal target for anti-EBV strategies. In addition to targeting the EBNA1 protein with small chemical inhibitors, targeting EBNA1 transcript levels has also been reported as a strategy for combating EBV and inhibiting EBV-associated tumors. Yin et al. demonstrated that using small interfering RNAs (siRNAs) can suppress EBNA1 expression. This RNA interference (RNAi) approach effectively reduces levels of endogenously expressed EBNA1 and inhibits the growth and survival of tumor cells in EBV-positive cell lines [54]. Hong et al. showed that targeting EBNA1 with small hairpin RNAs (shRNAs) significantly inhibited the proliferation of EBV-positive Raji cells [55]. Wang et al. also demonstrated that lentivirus-mediated RNA interference targeting the EBNA1 gene can inhibit the growth of xenografts composed of EBV-associated gastric carcinoma GT-38 cells in vivo [56]. Although RNAi-mediated targeting of EBNA1 has been shown to effectively suppress EBV-associated carcinoma both in vitro and in vivo, its clinical application faces certain limitations. RNAi-based approaches, including siRNAs, shRNAs, and locked nucleic acid antisense oligonucleotides (LNA-AOs), each have their drawbacks. Both siRNAs and LNA-AOs have relatively short half-lives within target cells and pose challenges in achieving efficient delivery to these cells in vivo [57], making them unsuitable for long-term gene therapy. Consequently, they are primarily used for research purposes rather than clinical applications. On the other hand, shRNAs can be expressed from viral vectors, allowing for effective delivery and sustained expression. However, the excessive expression of shRNAs can saturate the Argonaute/RNA-induced silencing complex (RISC), potentially disrupting the function of cellular microRNAs (miRNAs) [58], which makes shRNA-based methods less satisfactory for gene therapy. In contrast, the CRISPR/Cas13 system is well-suited to the demands of gene therapy. Unlike RNAi-based methods, CRISPR/Cas13 does not saturate cellular processes because it originates from bacteria, and humans lack components that utilize a similar mechanism. Therefore, it does not interfere with cellular small RNA pathways. Additionally, similar to shRNA, CRISPR/Cas13 can be delivered using viral vectors, particularly AAV. The targeting can be fine-tuned by selecting appropriate AAV serotypes. For instance, we have demonstrated that AAV D/J is likely the most suitable serotype for delivering CRISPR/Cas13 to nasopharyngeal cells, as it exhibits high transduction efficiency in both normal nasopharyngeal epithelial cells and transformed nasopharyngeal carcinoma cells. This capability allows for the delivery of CRISPR/Cas13 constructs in both pre-cancerous and cancerous states, expanding the potential therapeutic applications for patients. Consequently, CRISPR/Cas13 holds significant promise for targeting viral transcripts, and our study provides a proof-of-concept for its application in anti-EBV strategies.

Many studies have demonstrated the potential of CRISPR/Cas13 for antiviral RNA therapies. However, our research is the first to explore its use against a DNA virus. Since DNA viruses can be directly targeted using DNA editing techniques, CRISPR/Cas9 is often the preferred choice to explore against DNA viruses, including EBV. van Diemen et al. demonstrated that using a lentivirus with a single gRNA to target EBNA1 and OriP could deplete 40–60% of the EBV genome in EBV-positive Burkitt’s lymphoma Akata-Bx1 cells, with efficiency reaching up to 90% using double RNAs targeting the EBNA1 region [59]. Our team has also reported that CRISPR/Cas9 can reduce the EBV genome by 50% in NPC cells C666-1 by targeting EBNA1 and OriP, although further enhancement with additional gRNAs was not successful [18]. This discrepancy may be primarily due to the differing EBV copy numbers between B cells and NPC cells. Akata/BX1 cells are known to have about 10 copies of the EBV episome per cell [60], whereas C666-1 cells contain more than 50 copies per cell [61]. This higher copy number in NPC cells makes it more challenging to completely eliminate EBV genomes compared to B cells. Additionally, a greater number of viral copies increases the likelihood of EBV clones developing mutations at the edited sites, allowing them to evade double cleavage and elimination by CRISPR/Cas9. Therefore, we need to explore additional strategies to enhance the effectiveness of DNA editing approaches for eliminating EBV in NPC cells. CRISPR/Cas13 strategies are well-suited for this purpose. Although CRISPR/Cas13 shows some effect, albeit not very potent, in eliminating the EBV genome (Figure 3A), it has significant potential as an adjunct to current treatments. Our findings demonstrate its additive effect on CRISPR/Cas9-mediated anti-EBV strategies (Figure 3B) and chemotherapy (Figure 5). Since both CRISPR/Cas9 and CRISPR/Cas13 can be expressed and delivered via viral vectors, they can be packaged into AAV vectors for co-delivery into target cells [62]. This is particularly effective when using AAV D/J, which we identified as more efficient for nasopharyngeal epithelial cells. CRISPR/Cas9 provides the primary mechanism for damaging the EBV genome, while CRISPR/Cas13 reduces the EBNA1 protein levels, aiding in the elimination of the damaged EBV episome. Additionally, CRISPR/Cas13, with or without CRISPR/Cas9 co-treatment, increases the sensitivity of NPC cells to chemotherapy. This approach enables for a reduction in the effective dosage of chemotherapeutic drugs, minimizing their non-specific side effects on patients.

The application of CRISPR/Cas13 as an adjuvant could benefit from the approach used in nasal spray vaccines for prophylactic purposes. Nasal spray vaccines are administered through the nostrils, specifically targeting the mucous membranes of the respiratory tract. Often formulated as live attenuated vaccines, they can provide enhanced protection against respiratory infections. FluMist is a well-known example, used to guard against influenza [63]. Following the COVID-19 pandemic, numerous groups and companies have been working on developing nasal spray vaccines for SARS-CoV-2 [64], contributing to the stabilization and maturation of this technology for human use. Given the potential of CRISPR/Cas13-mediated EBV suppression to reduce EBV load in NPE, it might be worthwhile to explore whether prophylactic administration of CRISPR/Cas13 constructs via recombinant AAV could help prevent the development of NPC, similar to the preventive use of nasal spray vaccines. Additionally, this application could be considered for patients diagnosed with NPC, to be used prior to their chemotherapy treatment. Since the nasopharynx is located in the upper respiratory tract, recombinant AAVs can be easily delivered to this target site. With the appropriate AAV serotype, CRISPR/Cas constructs can be effectively delivered to the nasopharyngeal epithelial cells to perform their function. This approach could potentially create a barrier against EBV infection by continuously expressing CRISPR/Cas13 to suppress the EBNA1 transcript in normal NPE or precancerous cells. Overall, our study highlights the potential of CRISPR/Cas13-mediated EBV suppression in NPC therapy and creates new possibilities for exploring the prophylactic prevention of NPC development.

## 5. Conclusions

In summary, our findings demonstrate the potential of CRISPR/Cas13-mediated EBNA1 inhibition as a promising therapeutic strategy against EBV-associated NPC. This approach not only effectively reduces viral load but also acts as an adjuvant, enhancing the efficacy of existing chemotherapeutic treatments and CRISPR/Cas9-mediated EBV genome editing. By sensitizing NPC cells to conventional therapies and improving the targeting of the EBV genome, CRISPR/Cas13 offers a multifaceted approach to combating this challenging malignancy. Further research, including in vivo studies and clinical trials, is warranted to fully explore and optimize the translational potential of CRISPR/Cas13, alone or in combination with other modalities, for the treatment of NPC and other EBV-related malignancies.

## Figures and Tables

**Figure 1 viruses-17-00899-f001:**
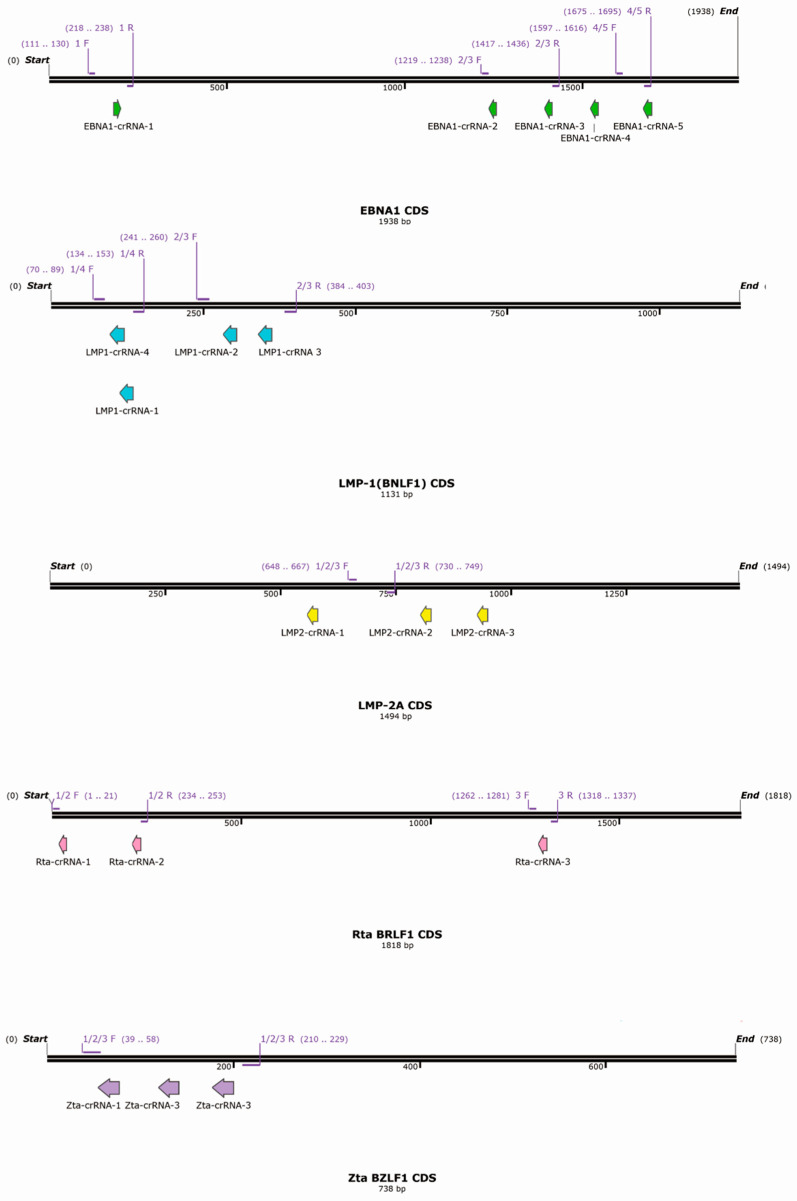
crRNA design. A schematic diagram illustrates the target sites for CRISPR/Cas13 editing within the coding regions of EBNA1, LMP1, LMP2, Rta, and Zta. Arrows highlight the crRNA target sites. Additionally, the diagram shows the binding sites of specific primers used to assess knockdown efficiency. The diagram was prepared using SnapGene^®^ Viewer v4.1.

**Figure 2 viruses-17-00899-f002:**
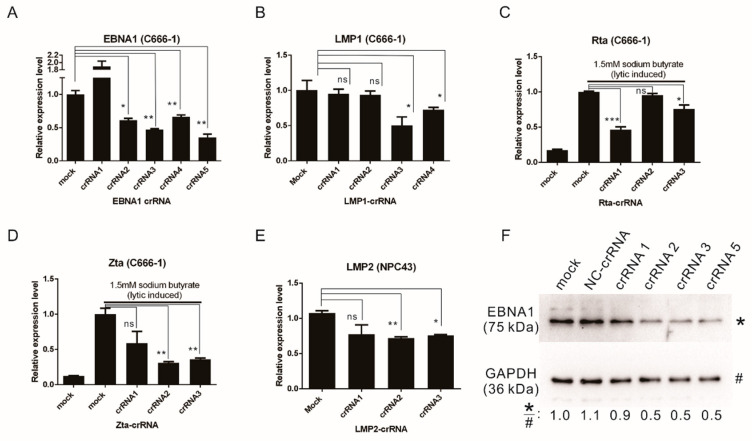
CRISPR/Cas13-mediated suppression of EBV transcripts in nasopharyngeal carcinoma cells. (**A**,**B**) The knockdown efficiency of CRISPR/Cas13-mediated suppression of EBNA1 and LMP1 was tested in C666-1 cells. A total of 1 × 10^5^ C666-1 cells were transfected with two rounds of pXR001 and pXR003-EBNA1/LMP1 crRNA, using a CasRx (pXR001)-to-crRNA (pXR003) ratio of 1:5. Transfections were performed on days 1 and 3, and total RNA was extracted 5 days post-transfection. The EBNA1 and LMP1 transcripts were analyzed by qRT-PCR and the level of EBNA1 mRNA was relative to that of Tubulin mRNA. (**C**,**D**) The knockdown efficiency of CRISPR/Cas13-mediated suppression of Rta and Zta was assessed upon lytic induction in C666-1 cells. The cells were transfected with two rounds of pXR001 and pXR003-Rta/Zta crRNA on days 1 and 3. On day 4, the transfected cells were treated with 1.5 mM sodium butyrate for lytic induction, and total RNA was extracted 5 days post-transfection for qRT-PCR analysis. (**E**) The knockdown efficiency of CRISPR/Cas13-mediated suppression of LMP2 was tested in NPC 43 cells. A total of 0.8 × 10^5^ NPC 43 cells were transfected with two rounds of pXR001 and pXR003-LMP2 crRNA on days 1 and 3, and total RNA was extracted 5 days post-transfection for analysis. Bars represent the means of 3 biological replicates, with error bars indicating standard deviation (SD). Asterisks indicate the significance of the difference between the compared groups (ns: *p* > 0.05; *: *p* < 0.05; **: *p* < 0.01; ***: *p* < 0.001). (**F**) Western blot analysis of EBNA1 proteins in EBNA1-crRNA transfected cells was performed. The relative expression levels of the EBNA1 protein were analyzed using Image J (version 1.54p) and are illustrated below the diagram.

**Figure 3 viruses-17-00899-f003:**
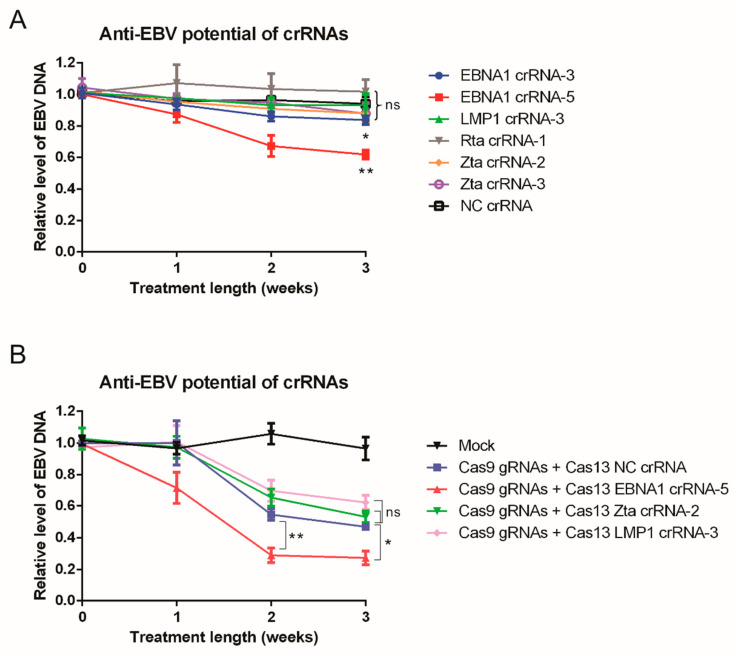
CRISPR/Cas13-mediated suppression of EBV DNA load in C666-1 cells. (**A**) Evaluation of the anti-EBV potential of selected crRNAs. A total of 1 × 10^5^ C666-1 cells in a 6-well plate were co-transfected on Day 0 with pXR001 and pXR003-EBV-crRNA at a ratio of 1:5. Every three days post-transfection, the transfected cells were trypsinized and re-seeded into a new plate with 1 × 10^5^ cells, followed by the same transfection on the next day. This process was repeated and lasted for approximately three weeks. A non-relevant crRNA (NC-crRNA) was used as the negative control. Total genomic DNA was harvested every week after the first transfection, and the EBV DNA load was determined by TaqMan qPCR. The differences in EBV DNA load between day 0 and four weeks post-transfection for EBNA-1-crRNA-3 (*p* = 0.01) and EBNA-1-crRNA-5 (*p* = 0.003) were statistically significant, as determined by Student’s *t*-test. (**B**) Synergistic anti-EBV strategies combining CRISPR/Cas9 and CRISPR/Cas13. C666-1 cells were co-transfected with the gRNA-Cas9 co-expression plasmids (PX459-EBNA1-gRNA1, PX459-EBNA1-gRNA-2, and PX459-OriP-gRNA-1) and the CRISPR/Cas13 expression constructs (pXR001 and pXR003-EBV-crRNA) at a 1:4 ratio. The transfection procedure followed the description in (**A**) with a transfection interval of 3 days. Total genomic DNA was harvested every week after the first transfection, and the EBV DNA load was determined by TaqMan qPCR. The differences between the Cas13 NC crRNA and the Cas13 EBNA1-crRNA-5 groups were statistically significant in week 3 (*p* = 0.006) and week 4 (*p* = 0.017) after the first transfection, as determined by Student’s *t*-test (ns: *p* > 0.05; *: *p* < 0.05; **: *p* < 0.01).

**Figure 4 viruses-17-00899-f004:**
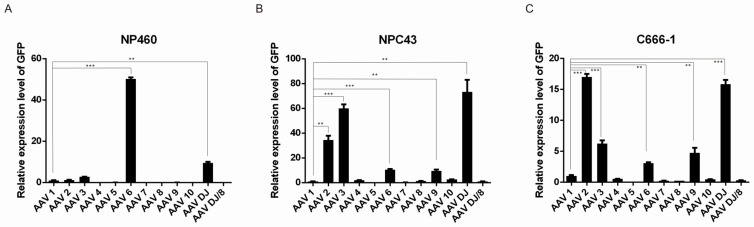
Transduction activity of different AAV serotypes to NPE and NPC cells. The transduction efficiency of various AAV serotypes in the NPE cell line NP460 and the NPC cell lines C666-1 and NPC43 was evaluated using twelve pre-made GFP-expressing AAV serotypes (1, 2, 3, 4, 5, 6, 7, 8, 9, 10, DJ, and DJ/8) at an MOI of 1:10,000. The relative GFP expression levels were measured 48 h post-transduction in the transduced cells—NP460 (**A**), NPC43 (**B**), and C666-1 (**C**)—were measured using qRT-PCR. The GFP mRNA levels were normalized to the tubulin mRNA levels, and the fold change was calculated relative to AAV1, which was set as the reference with a transduction efficiency of 1 Significant differences in transduction efficiency compared to AAV1 were observed with AAV 6 (*p* = 0.0001) and AAV DJ (*p* = 0.002) in NP460; AAV 2 (*p* = 0.003), AAV 3 (*p* = 0.001), AAV 6 (*p* = 0.001), AAV 9 (*p* = 0.005), and AAV DJ (*p* = 0.006) in NPC43; and AAV 2 (*p* = 0.0001), AAV 3 (*p* = 0.001), AAV 6 (*p* = 0.005), AAV 9 (*p* = 0.009), and AAV DJ (*p* = 0.0004) in C666-1 cells (**: *p* < 0.01; ***: *p* < 0.001).

**Figure 5 viruses-17-00899-f005:**
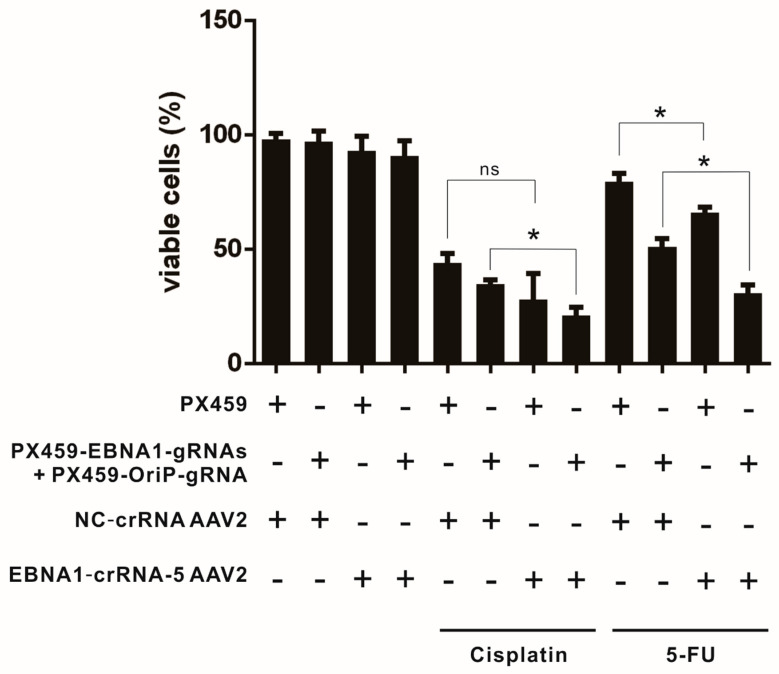
CRISPR/Cas13 targeting of EBNA1 enhances chemotherapeutic sensitivity in C666-1 cells. Cells were initially transduced with either EBNA1-crRNA-5/CasRx co-expressing AAV2 (EBNA1-crRNA-5 AAV2) or NC-crRNA/CasRx co-expressing AAV2 (NC-crRNA AAV2) on day 0. Three days after transduction, the cells were re-seeded and transfected with CRISPR/Cas9 constructs targeting the EBNA1 and OriP regions (PX459-EBNA1-gRNAs + PX459-OriP gRNA) or a mock vector (PX459). This transfection process was repeated every three days. Two weeks after the initial transduction, the cells underwent another round of AAV transduction followed by multiple transfections. The entire treatment spanned four weeks. After the treatment period, 5 × 10^4^ C666-1 cells were seeded in a 96-well plate with 200 μL of growth medium, one day prior to conducting the MTS assay. The cells were then treated with 0.1 mM cisplatin for 4 h or 1 μM 5-fluorouracil (5-FU) for 24 h. Cell viability was assessed using the MTS assay. Statistically significant differences were observed between the mock-transduced and EBNA1-crRNA-5 AAV2-transduced groups treated with 5-fluorouracil (*p* = 0.046), as well as between the CRISPR/Cas9 single-treated and CRISPR/Cas9 and Cas13 co-treated groups when treated with cisplatin (*p* = 0.041) or 5-fluorouracil (*p* = 0.003), as determined by Student’s *t*-test (ns: *p* > 0.05; *: *p* < 0.05).

## Data Availability

Data is contained within the article or Appendix A.

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
