# Peer review of "CRISPR/Cas13-Mediated Inhibition of EBNA1 for Suppression of Epstein–Barr Virus Transcripts and DNA Load in Nasopharyngeal Carcinoma Cells"

_viruses, 2025, doi:10.3390/v17070899_

Round 1
Reviewer 1 Report
Comments and Suggestions for Authors
This is an excellent and thorough experimental study that explores the potential of CRISPR/Cas13 as a therapeutic approach against EBV.
Minor revision is needed for publication.
Please add keywords.
Panel A (Figure 4) does not provide much visual information (all blocks except one appear black). This panel could be moved to Supplementary data.
Section 2.7 describes the MTS assay as a proliferation assay. However, Figure 5 shows data as "viable cells", suggesting a cytotoxic effect but not a proliferative effect. Antiproliferative and cytotoxic effects are different by definition. Please correct the description of this method.
A Conclusion section is strongly recommended.
Author Response
Thank you for your comments and support regarding the manuscript. We have addressed your suggestions as follows:
Comments 1: Please add keywords.
Response 1: Thank you for bringing this to our attention. We have added four keywords to the manuscript. Line 25 "Keywords: Epstein-Barr virus; Nasopharyngeal carcinoma; CRISPR/Cas13; EBNA1 Inhibition".
Comments 2: Panel A (Figure 4) does not provide much visual information (all blocks except one appear black). This panel could be moved to Supplementary data.
Response 2: We agree with this comment and have moved Panel A to Supplementary Figure 1.
Comments 3: Section 2.7 describes the MTS assay as a proliferation assay. However, Figure 5 shows data as "viable cells", suggesting a cytotoxic effect but not a proliferative effect. Antiproliferative and cytotoxic effects are different by definition. Please correct the description of this method.
Response 3: Thank you for pointing this out. We have corrected the description in Section 2.7 to "Cell Viability Assay."
Comments 4: A Conclusion section is strongly recommended.
Response 4: We apologize for omitting the conclusion. The conclusion section has now been added to the manuscript.
Reviewer 2 Report
Comments and Suggestions for Authors
“CRISPR/Cas13-Mediated Inhibition of EBNA1 for Suppression of Epstein-Barr Virus Transcripts and DNA Load in Nasopharyngeal Carcinoma Cells” manuscript explores the potential of CRISPR/CRISPR-associated protein 13 (Cas13) in combination with CRISPR/Cas9 to eliminate EBV virus. The manuscript reports the treatment of pharyngeal carcinoma (NPC) and found that cotreatment of NPC increased cell sensitivity to 5-fluorouracil and cisplatin. This research is important especially that noncommunicable diseases continue spreading, causing health burden worldwide and putting millions of individuals at risk. The research will help in the discovery of effective therapy to eliminate NPC. The manuscript is clear and follows the idea from the beginning to the end. The references used in the manuscript are relevant to the topic of discussions and are up to date. However, there are few comments to the authors that should be addressed before the manuscript is published.
Broader comments
It could improve reading if the authors could state clearly why the CRISPR/CRISPR 13 is been explored for targeting DNA virus (EBV). The clear novelty of using CRISPR/CRISPR 13 for EBV at the end of the introduction could be stated to improve the readability. The use of passive language in the manuscript should revised and the authors should go through the manuscript to address the wording problem as I have mentioned bellow.
Minor comments
Lines 17-18: phrase enhanced elimination of the virus could be more direct “for example enhanced viral clearance etc
Line 50: typo “smolecule inhibitors”, I think it should read small molecule inhibitors
Line 63: to enhance the readability….the sentence could ready like “our previous work demonstrated”
Author Response
Thank you for your comments and support regarding the manuscript. We have addressed your suggestions as follows:
Comments 1: It could improve reading if the authors could state clearly why the CRISPR/CRISPR 13 is been explored for targeting DNA virus (EBV).
Response 1: Thank you for highlighting this issue. The advantages of using CRISPR/Cas13 on DNA viruses have now been added to the second-to-last paragraph of the introduction. The updated sections are highlighted in red from lines 106 to 114.
Comments 2: The clear novelty of using CRISPR/CRISPR 13 for EBV at the end of the introduction could be stated to improve the readability.
Response 2: Thank you for pointing this out. The novelty of using CRISPR/Cas13 for EBV has now been added to the end of the introduction. The updated sections are highlighted in red from lines 126 to 128.
Comments 3: The use of passive language in the manuscript should revised.
Response 3: We apologize for using too much passive language in the manuscript. We have rephrased some sentences to adopt a more active format. The updated sections are highlighted in red on lines 82, 569-570, 623-624, 851, and 656-657.
Comments 4: address the wording problem as I have mentioned bellow.
Lines 17-18: phrase enhanced elimination of the virus could be more direct “for example enhanced viral clearance etc
Line 50: typo “smolecule inhibitors”, I think it should read small molecule inhibitors
Line 63: to enhance the readability….the sentence could ready like “our previous work demonstrated”
Response 4: Thank you for pointing out the wording issues and the typo. We have made corrections accordingly based on your three suggestions.
Reviewer 3 Report
Comments and Suggestions for Authors
The authors demonstrated that the CRISPR/Cas13 system is promising for suppression of RNA virus replication using EBV as an example. CRISPR/Cas13 can effectively suppress target transcripts in human cells. Knockdown of RNA virus transcript may be used in therapy as an adjunct to existing treatments.
Some comments are listed below:
Line 27. Please add information about the mechanisms of nasopharyngeal cancer development after EBV infection.
Line 30-33 “NPC-associated EBV infections are considered…” this sentence is not clear
Line 284. Please describe the process of developing crRNAs targeting EBV transcripts. This should be added to the Methods section
Line 291. Please add a link to the online BLAST tool. The method for analyzing crRNAs should also be added to the Methods section. Line 302. Is SnapGene® Viewer an online tool? If so, please provide a link.
Line 305. Why was a 1:5 ratio chosen?
Line 331 and 356. This is a duplicate
Line 357. “approximately”. Please provide more specific information.
Line 407. Why was a 1:4 ratio chosen?
Section 3.4. Please specify what was chosen as a control. This should also include a comparison of vector and non-vector delivery efficiency.
Figure 4 B-D. Please provide information on the significance of the differences.
Author Response
Thank you for your comments and support regarding the manuscript. We have addressed your suggestions as follows:
Comments 1: Line 27. Please add information about the mechanisms of nasopharyngeal cancer development after EBV infection.
Response 1: Thank you for highlighting this issue. We have now reorganized the first paragraph of the introduction to emphasize the importance of EBV's role in NPC development, even though the detailed oncogenesis mechanism remains undetermined. The updated sections are highlighted in red from lines 32 to 38.
Comments 2: Line 30-33 “NPC-associated EBV infections are considered…” this sentence is not clear
Response 2: Apologies for the lack of clarity. The paragraph has been rewritten along with the changes mentioned in response 1.
Comments 3: Line 284. Please describe the process of developing crRNAs targeting EBV transcripts. This should be added to the Methods section
Response 3: Thank you for highlighting this. The process of developing crRNAs has now been added to the methodology in Section 2.1, "Plasmid and crRNA Design." The updated sections are highlighted in red from lines 134 to 139.
Comments 4: Line 291. Please add a link to the online BLAST tool. The method for analyzing crRNAs should also be added to the Methods section. Line 302. Is SnapGene® Viewer an online tool? If so, please provide a link.
Response 4: We agree with this comment. The web links for BLAST and RNAfold have been added to the text (lines 304 and 312). The analysis of the crRNAs has also been included in Section 2.1, "Plasmid and crRNA Design," of the methodology. Since SnapGene® Viewer is a software application and not an online tool, no web link has been added for it.
Comments 5: Line 305. Why was a 1:5 ratio chosen?
Response 5: In our preliminary trials (data not shown), we tested different ratios of 8:1, 1:1, and 1:5, and found that the 1:5 ratio provided the best suppressive effect. Therefore, we chose the 1:5 ratio for our study.
Comments 6: Line 331 and 356. This is a duplicate
Response 6: Thank you for pointing this out. The sentence starting on line 331 has now been deleted.
Comments 7: Line 357. “approximately”. Please provide more specific information.
Response 7: We apologize for the unclear content. The statement has been revised to specify "transfection for 5 days" instead of "approximately a week."
Comments 8: Line 407. Why was a 1:4 ratio chosen?
Response 8: As in response 5, the ratio used was determined based on our preliminary trials.
Comments 9: Section 3.4. Please specify what was chosen as a control. This should also include a comparison of vector and non-vector delivery efficiency.
Response 9: Thank you for highlighting this. We calculated the fold change relative to AAV serotype 1 transduction. Since the GFP-expressing AAV serotypes were pre-made, we lack a non-vector control. Therefore, we compared results to AAV serotype 1 because it has relatively low transduction efficiency in NPE and NPC cells. The text has been updated on lines 495 to 496 to specify the comparison method.
Comments 10: Figure 4 B-D. Please provide information on the significance of the differences.
Response 10: Apologies for omitting the statistical analysis for Figure 4. A Student's t-test analysis has now been added to the figure and its legend.